# Immune Reconstitution Inflammatory Syndrome Induced by *Mycobacterium avium* Complex Infection Presenting as Chronic Inflammatory Demyelinating Polyneuropathy in a Young AIDS Patient

**DOI:** 10.3390/medicina58010110

**Published:** 2022-01-11

**Authors:** An-Che Cheng, Te-Yu Lin, Ning-Chi Wang

**Affiliations:** 1Department of Medicine, Tri-Service General Hospital, National Defense Medical Center, Taipei City 11490, Taiwan; kwii781215@gmail.com; 2Division of Infectious Disease and Tropical Medicine, Department of Medicine, Tri-Service General Hospital, National Defense Medical Center, Taipei City 11490, Taiwan; lin.deyu@msa.hinet.net

**Keywords:** HIV, immune reconstitution inflammatory syndrome, *Mycobacterium avium* complex, chronic inflammatory demyelinating polyneuropathy

## Abstract

Antiretroviral therapy (ART) can restore protective immune responses against opportunistic infections (OIs) and reduce mortality in patients with human immunodeficiency virus (HIV) infections. Some patients treated with ART may develop immune reconstitution inflammatory syndrome (IRIS). *Mycobacterium avium* complex (MAC)-related IRIS most commonly presents as lymphadenitis, soft-tissue abscesses, and deteriorating lung infiltrates. However, neurological presentations of IRIS induced by MAC have been rarely described. We report the case of a 31-year-old man with an HIV infection. He developed productive cough and chronic inflammatory demyelinating polyneuropathy (CIDP) three months after the initiation of ART. He experienced an excellent virological and immunological response. Sputum culture grew MAC. The patient was diagnosed with MAC-related IRIS presenting as CIDP, based on his history and laboratory, radiologic, and electrophysiological findings. *Results*: Neurological symptoms improved after plasmapheresis and intravenous immunoglobulin (IVIG) treatment. To our knowledge, this is the first reported case of CIDP due to MAC-related IRIS. Clinicians should consider MAC-related IRIS in the differential diagnosis of CIDP in patients with HIV infections following the initiation of ART.

## 1. Introduction

Antiretroviral therapy (ART) dramatically decreases plasma viral load and improves CD4 T cell counts in patients with human immunodeficiency virus (HIV) infections. These immunological changes correlate with decreases in the frequency of opportunistic infections (OIs) and reductions in morbidity and mortality rates [1]. However, some patients treated with ART may develop immune reconstitution inflammatory syndrome (IRIS), which is characterized by paradoxical clinical worsening of treated OIs or unmasking of previously subclinical untreated infections.

*Mycobacterium avium* complex (MAC)-related IRIS most commonly presents as lymphadenitis, soft-tissue abscesses, and deteriorating lung infiltrates [2]. Nonetheless, the neurological manifestations of MAC-related IRIS have rarely been described [3,4,5,6]. Herein, we report the case of a patient, with an HIV infection, who presented with chronic inflammatory demyelinating polyneuropathy (CIDP) due to unmasking MAC-related IRIS. To the best of our knowledge, this is the first report to discuss this unusual presentation.

## 2. Case Presentation

A 31-year-old man, who was in good health until 26 October 2018, presented to our outpatient department with a weight loss of 8 kg (from 75 kg to 63 kg), which had occurred over the preceding month. He was diagnosed with HIV infection. Immunology revealed lymphocytopenia with an absolute lymphocyte count of 373 cells/μL, CD4 T cell count of 4 cells/μL, CD8 T cell count of 294 cells/μL, and CD8/CD4 ratio to 73.5. His plasma HIV RNA load was 586,300 copies/mL (log value: 5.77) (Figure 1). Results of blood cultures and fungal antigen testing were negative for bacterial, mycobacterial, and fungal pathogens. He had no respiratory symptoms, no cough, and was unable to produce induced sputum specimen. Chest radiography showed no abnormality. ART was initiated on 2 November 2018, and the following drugs were administered daily: elvitegravir (150 mg), cobicistat (150 mg), emtricitabine (200 mg), and tenofovir alafenamide (10 mg). Four weeks after the initiation of ART, his absolute lymphocyte count increased to 943 cells/μL, CD4 T cell count to 98 cells/μL, CD8 T cell count to 466 cells/μL, CD8/CD4 ratio to 4.75, and his HIV RNA load had decreased significantly to 62 copies/mL (log value: 1.8) (Figure 1). The antiretroviral regimen was well-tolerated, and he experienced no adverse effects.

Three months after ART was initiated, the patient visited our emergency department due to productive cough, rapidly progressive quadriparesis, and lower limb paresthesia, which he had been experiencing for 3 days. Examination revealed vital signs within the normal range (body temperature: 36.5 °C; pulse: 83 beats/min; blood pressure: 121/77 mmHg; respiratory rate: 18 breaths/min) and an arterial oxygen saturation of 97% while breathing ambient air. He exhibited both proximal and distal quadriparesis with 2/5 muscle power in the left upper limb and 3/5 muscle power in the remaining limbs. Hyporeflexia was observed in the bilateral lower limbs above the knee and ankle, while the sensory loss was observed below the knee. Meningeal signs were absent.

Laboratory examinations revealed a white blood cell count of 8260 cells/µL and a C-reactive protein level of 7.49 mg/dL. Platelet count, hemoglobin, serum sodium, potassium, free calcium, magnesium, coagulation, and renal and liver function tests all yielded results within the normal range. His absolute lymphocyte count increased to 1734 cells/μL, CD4 T cell count to 109 cells/μL, CD8 T cell count to 726 cells/μL, CD8/CD4 ratio to 6.66, and his viral load was 113 copies/mL (log value: 2.1) (Figure 1). The serum IgA, IgG, and IgM in the patient were within normal range. No autoantibody against peripheral nerve antigens was detected in the serum of the patient. The toxicology screen, thyroid function tests, Lyme titers, and vasculitis markers showed no abnormalities. Cerebrospinal fluid (CSF) analysis revealed the following: clear appearance, positive Pandy’s test, white blood cell count of 1/µL, red blood cell count of 26/µL, total proteins level of 63 mg/dL, and glucose level of 52 mg/dL, which suggested albuminocytologic dissociation. CSF tests for syphilis, cryptococcus, streptococcus, meningococcus, pneumococcus, *H. influenzae B*, measles, mycobacteria, cytomegalovirus, Epstein–Barr virus, herpes simplex 1 and 2, and enterovirus varicella zoster virus were negative. Sputum cultures were positive for MAC.

Chest radiography showed no abnormality. Electromyography (EMG) and nerve conduction velocity (NCV) studies revealed a prolonged F-wave latency of 40% above the upper limit of normal in the left median and left ulnar nerves, a 30% reduction in the amplitude of the left median and ulnar compound muscle action potentials, and a 20% decrease in conduction velocity in the left median sensory nerve, which suggested a demyelinating polyneuropathy with axonal damage. The magnetic resonance imaging (MRI) of the cervical and lumbar spine revealed no significant abnormality. Brain MRI revealed high signal intensities over both posterior corona radiata and centrum semiovale (Figure 2).

The patient was initially diagnosed with unmasking MAC-related IRIS with acute inflammatory demyelinating polyneuropathy (AIDP). After 10 cycles of plasma exchange and treatment with clarithromycin (500 mg) and ethambutol (1200 mg), gradual improvement in neurological symptoms was observed. He exhibited 4/5 muscle power in the left upper limb, both proximally and distally, and 5/5 muscle power in the other limbs, both proximally and distally. However, the disease relapsed two months later, with 3/5 muscle power in the left upper limb, proximally and distally, and 4/5 muscle power in the other limbs, proximally and distally. After 5 days of treatment with intravenous immunoglobulin (IVIG), his muscle power had returned to normal in all limbs, with no signs of residual sensory loss or hyporeflexia. The patient was finally diagnosed with CIDP caused by unmasking MAC-related IRIS.

## 3. Discussion

IRIS is a clinical condition characterized by excessive inflammatory response to a preexisting antigen or pathogen and paradoxical deterioration in clinical status after initiation of ART [7]. Although IRIS is a well-known phenomenon, there is still no universally accepted definition. However, Robertson et al. [8] proposed the following criteria: clinical signs and symptoms compatible with an inflammatory process, a temporal relationship between the initiation of ART and the development of symptoms, evidence of virologic response demonstrated by a decrease in plasma HIV RNA load of at least 1 log copy/mL. Furthermore, the clinical course should neither be consistent with the usual course of a previously diagnosed opportunistic infection nor a new infectious process, and the signs and symptoms should not be explained by drug toxicity. Although an increase in CD4 T cell count is not a required criterion, it provides supporting evidence of the diagnosis of IRIS. The International Network for the Study of HIV-associated IRIS (INSHI) [9] defined two categories of IRIS: (1) paradoxical reaction after the start of ART in patients receiving OIs treatment (termed paradoxical IRIS), or (2) the “unmasking” of OIs that were not clinically apparent prior to ART, as these develop to become clinically recognizable after the initiation of ART because of the restoration of antigen-specific functional immune responses (termed unmasking IRIS). INSHI [9] proposed the diagnostic criteria for unmasking IRIS as follows: (1) the patient is not receiving treatment for OIs when ART is initiated; (2) a new onset of symptoms and diagnosis of OIs is found within 3 months following the initiation of ART; and (3) there is evidence of a marked inflammatory component to the presentation. In this report, the patient displayed no evidence of OIs and no respiratory symptoms before ART started. MAC prophylaxis was not given. After initiating ART, he experienced a good virologic response with a decrease in plasma viral load, and a restoration of his immune function with an increase in CD4 T cell count. However, he presented with a productive cough and neurological symptoms and signs 3 months after initiation of ART despite adequate immunity. The sputum culture results were subsequently positive for MAC. There was no evidence of central nervous system infection. The clinical presentations, temporal relationship between ART initiation and onset of symptoms, and microbiological findings in the patient had fulfilled the diagnostic criteria of unmasking MAC-related IRIS. The diagnosis of CIDP was established based on the clinical manifestations, immunology, CSF cytology, imaging, EMG, and NCV findings. Previous studies reported an aberrant immune response during immune reconstruction, after the initiation of ART, which may lead to inflammatory peripheral neuropathy [10,11,12]. The patient was finally diagnosed with CIDP due to unmasking MAC-related IRIS.

The major differential diagnoses included multifocal motor neuropathy (MMN), distal acquired demyelinating symmetric neuropathy (DADS) with monoclonal IgM gammopathy, and anti-myelin-associated glycoprotein antibodies (anti-MAG), and chronic ataxic neuropathy with ophthalmoplegia, M-protein, cold agglutinins, and disialosyl antibodies (CANOMAD). MMN is a chronic, immune-mediated neuropathy with asymmetric, predominantly distal often upper limb weakness in the absence of objective sensory involvement [13]. Anti-GM1 IgM antibodies have been reported with varying prevalence in patients with MMN ranging from 40–50% [14,15]. MMN usually responds to IVIG but not to plasma exchange [16]. DADS associated with IgM gammopathy typically has a slowly progressive, distal, predominantly sensory phenotype [17]. More than 50% of patients with an IgM paraprotein have anti-MAG IgM antibodies [18]. The presence of high titers of anti-MAG antibodies excludes the diagnosis of CIDP [19]. CANOMAD is a disorder with clinical features consisting of severe sensory ataxia and cranial nerve involvement including ophthalmoplegia, dysphagia, or dysarthria and only minimal weakness [20]. CANOMAD typically progresses over the years and peripheral neuropathy may precede the development of other features such as ophthalmoplegia [21]. In our case, the patient presented a rapidly progressive quadriparesis, with sensory involvement, and without symptoms of ophthalmoplegia, dysphagia, or dysarthria. Serum IgA, IgG, and IgM levels in the patient were within normal ranges, and no autoantibodies against peripheral nerve antigens were detected. The patient had a good response to IVIG. Based on the clinical scenario, laboratory immunological studies, treatment response, and MMN, DADS and CANOMAD were less likely.

The incidence of MAC-related IRIS is approximately 3.5% in patients with a baseline CD4 T cell count <100 cells/μL [22]. The common presentations of MAC-related IRIS include fever and lymphadenitis, followed by pulmonary disease [23]. Soft tissue nodules, osteomyelitis, and granulomatous hepatitis have also been reported in some cases [22]. However, neurological presentations of MAC-related IRIS have rarely been reported. To our knowledge, only four reports of MAC-related IRIS presenting with neurological complications have previously been published in English-language journals [3,4,5,6]. Previously reported neurological complications of MAC-related IRIS include meningoencephalitis, myelitis, multiple brain abscesses, and encephalitis. The symptoms and signs of the previous cases included fever, headache, disturbance of consciousness, aphasia, and paraplegia [3,4,5,6]. Every one of the previous cases presented with central nervous system involvement and worsening of a previously treated MAC infection after initiating ART (paradoxical MAC-related IRIS). However, in our case, the patient presented with progressive quadriparesis and peripheral nerve demyelination, and was diagnosed with CIDP due to a flare-up of an underlying, unmasking MAC-related IRIS. The median time patients were diagnosed with IRIS from ART initiation among previous cases and the present case was 17 months (range, 3–29 months). The median baseline CD4 T-cell count and plasma HIV RNA load was <10 cells/μL (range, 2–20 cells/µL) and 2.2 × 105 copies/mL (range, 1.7 × 104–5.9 × 105 copies/mL), respectively. Patients had a median CD4 T-cell count of 109 cells/μL (range, 10–210 cells/µL), and the plasma HIV RNA load decreased to <400 copies/mL at the time of IRIS diagnosis. The imaging findings of the brain in previously reported cases and our case were non-specific, and they included enhancement or ring enhancement of lesions, with or without focal edema and a mass effect. Three out of the four previously reported cases, and our case, had a favorable outcome. The clinical manifestations of the previously reported cases, and our case, are summarized in Table 1.

CIDP is an autoimmune peripheral neuropathy, induced by an aberrant immune response against self-antigens derived from peripheral nerves. Although the immunological mechanisms underlying the disease are not well understood, T cell-mediated and humoral immune mechanisms are considered to play an important role in the pathogenesis of CIDP [24]. During active disease, CD4 T cells secrete pro-inflammatory cytokines and chemokines into the circulation [25]. These cytokines and chemokines may help B cells to produce antibodies against peripheral nerves, and activate macrophages and cytotoxic CD8 T cells, contributing directly to the damage of both myelin and axons [26,27,28]. In recent years, autoantibodies against peripheral nerve antigens, namely neurofascin, contactin1, or contactin-associated protein 1, had been identified in CIDP [29,30]. These proteins are important in clustering Na^+^-channels and maintaining the functional structure of the myelinated axon which is essential for the saltatory conduction [31]. The incidence of these antibodies is reported at approximately 2–13% in CIDP patients [24,29,31]. The patients with these antibodies presented with a poor treatment response to IVIG in comparison to the patients who lack these antibodies in retrospective observations [32]. In this report, no autoantibodies against peripheral nerve antigens were detected, and the patient presented with a good treatment response to IVIG. The clinical presentations in our patient were consistent with findings in the previous study [32]. Testing these antibodies in CIDP patients may provide information for the selection of a suitable treatment approach for individual patients.

Although the pathophysiology of CIDP is understood in progress, limited information is available on the association between IRIS and CIDP. The relationship may be explained in part by lymphocyte proliferation, dysfunction of cytotoxic T lymphocyte-associated antigen-4 (CTLA-4), and regulatory T cell defects. Lymphocytopenia, especially CD4 T cell depletion, is common in patients with HIV infection. After initiating ART, the HIV viral load decreases, which enables lymphocyte proliferation. Lymphocyte proliferation may mediate the loss of control in immune homeostasis while compromising the mechanism protecting from self-reactivity, leading to an exacerbation of the immune response to self-antigens [33]. Moreover, impaired function of CTLA-4 is another hypothesized mechanism for immune reconstitution-associated autoimmune response in patients with HIV infection. CTLA-4 is a T cell-surface receptor that controls antigen-specific immunity and inhibits the immune response to autoantigens [34]. Some functional defects in CTLA-4 are associated with autoimmune disorders [35]. Patients, with an HIV infection, who have an initial CD4 T cell count <50 cells/µL and a substantial increase in CD4 T cell count after initiating ART, experience an increase in CTLA-4 production [36]. This may lead to a higher level of circulating soluble CTLA-4, which indirectly impairs CTLA-4 interaction with its ligands, including CD80 and CD86, on antigen-presenting cells [37,38]. Finally, regulatory T cells are the major regulators of the immune system homeostasis [39]. A loss of functional regulatory T cells leads to excessive T cell activation, and disproportionate inflammatory response [39]. Previous studies demonstrated that T regulatory cells were less effective in suppressing proliferative responses in patients with CIDP and IRIS than those from healthy controls [40,41,42]. Some authors found the numbers of circulating T regulatory cells to be reduced during the progressive phases of CIDP [41]. Our patient presented with lymphocytopenia and CD4 T cell depletion. He experienced an increase in lymphocyte and CD4 T cell count after initiating ART, which may indirectly impair function of CTLA-4 and lead to an uncontrolled immune response to self-antigens. However, we did not analyze the numbers and function of regulatory T-cells in this patient. Therefore, we could not determine the contribution of regulatory T cells to the pathogenesis in the present case. Further research needs to be conducted in order to understand the pathophysiologic mechanism of immune reconstitution-associated autoimmune neuropathy.

Although the peripheral nerve is targeted by an autoimmune attack, previous studies reported the infrequent coexistence of peripheral and central nervous system (CNS) demyelination [43,44,45]. Reported cases have shown the presence of white matter hyperintense T2 lesions on brain MRI in patients with CIDP, and these brain lesions commonly occur in multiples in the periventricular, subcortical, or brainstem white matter [43,45]. Previous case series suggested that CIDP patients with CNS involvement had a more favorable response to immunological treatment than those without CNS involvement [46]. The brain MRI in our case showed T2 and ADC map high signal intensities over both the posterior corona radiata and the centrum semiovale, which suggested white matter demyelination. In addition, the patient presented with a good treatment response to IVIG. The results of our study support the above notion.

Risk factors for IRIS have been investigated in previous studies. They include a low baseline CD4 T cell count (especially <50 cells/μL), higher baseline viral load, a rapid decrease in viral load within 3 months, and the presence of non-tuberculous mycobacteria and fungal OIs at the time of ART initiation [47,48]. However, the predictors of IRIS-associated autoimmune peripheral neuropathy, specifically CIDP, are unclear. Previous studies have shown an association between CD8 T cell counts and CIDP. In an animal model study conducted in mice, the CD8/CD4 ratio in the blood of diseased mice was approximately five times higher than that of healthy mice because of the marked increase in the absolute CD8 T cell count [49]. Matsumuro et al. [50] found that CD8 T cells outnumbered CD4 T cells at the sites of lesions in CIDP patients. Moreover, Mausberg et al. [51] found that CD8 T cells exhibited a broader clonal activation than CD4 T cells in the blood of CIDP patients. Our patient presented with an increased absolute CD8 T cell count and an abnormally high CD8/CD4 ratio 3 months after the initiation of ART (Figure 1), and subsequently developed CDIP. Although there are multiple immunopathological mechanisms involved in CIDP, we hypothesize that an increased CD8 T cell count with a high CD8/CD4 ratio may contribute to the development of CIDP. Further research is needed to identify predictors of CIDP.

## 4. Conclusions

We report the first case of unmasking MAC-related IRIS presenting with CIDP, and describe the clinical features, laboratory, CSF, and electrophysiological findings, and the treatment outcome. MAC-related IRIS should be considered in the differential diagnosis of CIDP in patients with HIV infection following the initiation of ART, and clinicians should carefully monitor patients who exhibit the aforementioned risk factors. Further studies should be conducted in order to gain a better understanding of the pathophysiologic mechanism of immune reconstitution associated with CIDP.

## Figures and Tables

**Figure 1 medicina-58-00110-f001:**
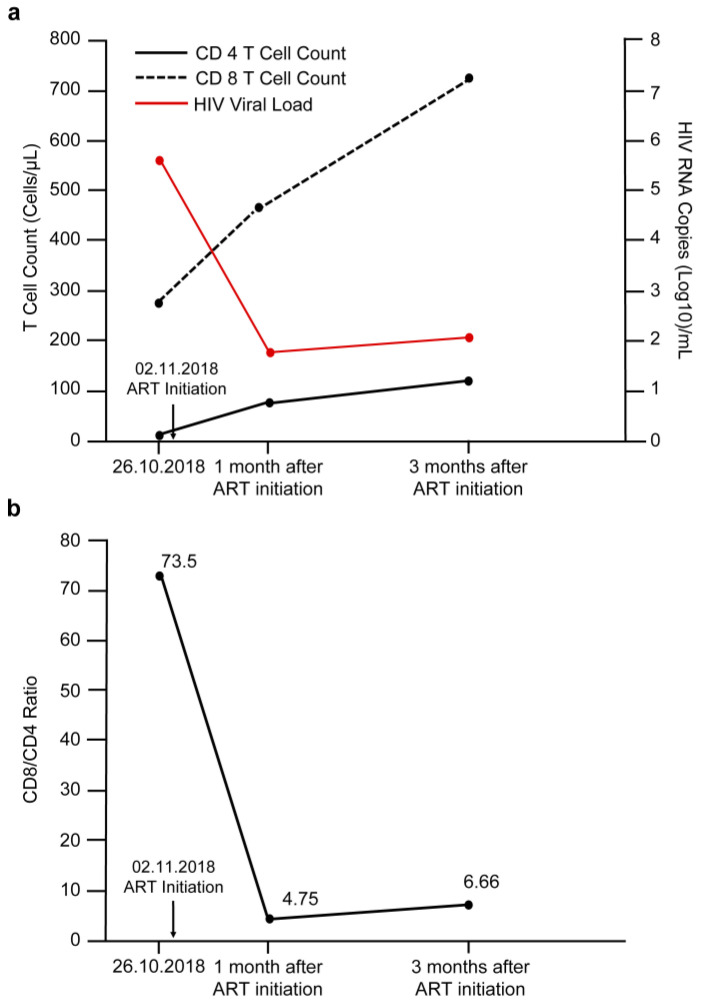
Kinetics of change in parameters include (**a**) CD4, CD8, HIV viral load, and (**b**) CD8/CD4 ratio at different time points.

**Figure 2 medicina-58-00110-f002:**
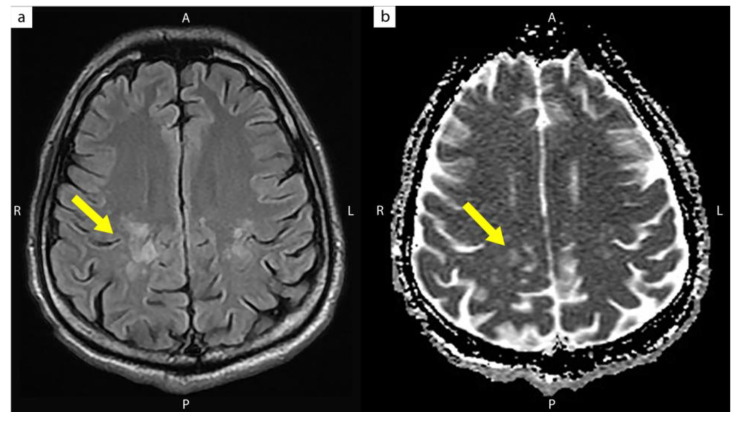
Brain magnetic resonance image without contrast showed (**a**) T2-weighted and (**b**) apparent diffusion coefficient (ADC) map high signal intensities over both the posterior corona radiata and centrum semiovale (yellow arrow).

**Table 1 medicina-58-00110-t001:** Case descriptions of MAC-related IRIS with neurological complications in relevant researches and the case in study.

Reference	Age/Sex	CD4 T-Cell Count at Baseline (Cells/μL)	CD4 T-Cell Count at IRIS Diagnosis (Cells/μL)	Plasma HIV RNA Load. at Baseline (Copies/mL)	Plasma HIV RNA Load. at IRIS Diagnosis (Copies/mL)	Time to IRIS Diagnosis from ART Initiation (Month)	MAC Related IRIS Presentation	Image Findings	Treatment	Outcome
[4]	24/M	2	70	77,600	<50	17 months	MAC meningoencephalitis and myelitis with drowsy, stumbling, and progressive paraplegia (Paradoxical IRIS)	Multiple enhancing nodules in the cerebral and cerebellar hemispheres, gray matter, brain stem and whole spinal cord.	Levofloxacin + clarithromycin ethambutol + rifabutin + dexamathasone	Significantly improved paraplegia and consciousness after 5 days of anti-MAC therapy
[5]	35/M	<10	210	382,987	<400	25 months	Cerebral MAC infection with headache, fever, dizziness, vomiting (Paradoxical IRIS)	A solitary 3 cm lesion in the left frontal lobe of brain with perifocal edema and mass effect	Rifabutin + isoniazid, Ethambutol + pyrazinamide + clarithromycin Excision surgery of brain	MAC-IRIS resolved 18 days after anti-MAC therapy
[6]	36/M	10	170	217,163	<50	29 months	Cerebral MAC abscesses with headache and aphasia (Paradoxical IRIS)	Two hypodense lesions with ring enhancement and edema in the right temporal lobe and left temporoparietal area	Azithromycin+ ethambutol + rifabutin	Complete regression of lesions 10 months after anti-MAC therapy
[3]	51/M	20	10	17,000	<50	3 months	Cerebral MAC infection with fever, disturbance of consciousness (Paradoxical IRIS)	Ring enhanced lesions with perifocal edema in the left temporal lobe	Ethambutol + isoniazid + levofloxacin + amikacin + dexamathasone	Died soon after onset of neurological symptoms
The present case	31/M	4	109	586,300	113	3 months	CIDP with progressive quadriparesis (Unmasking IRIS)	High signal intensities over both the posterior corona radiata and centrum semiovale	Ethambutol + clarithromycin Plasma exchange + IVIG	Significantly improved quadriparesis 3 months after anti-MAC and Plasma exchange + IVIG

IRIS: immune reconstitution inflammatory syndrome; ART: antiretroviral therapy; MAC: *Mycobacterium Avium Complex*; CIDP: chronic inflammatory demyelinating polyneuropathy; IVIG: intravenous immunoglobulin.

## Data Availability

The study did not report any data.

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
