# Peer review of "Immune Reconstitution Inflammatory Syndrome Induced by Mycobacterium avium Complex Infection Presenting as Chronic Inflammatory Demyelinating Polyneuropathy in a Young AIDS Patient"

_medicina, 2022, doi:10.3390/medicina58010110_

Round 1
Reviewer 1 Report
- The case presented for treatment in 2018 but was not submitted for publication until 2021, can you please explain.
- Authors did not report or discuss the role of regulatory T-cells frequency in this patient, which could play a role in the chronic inflammatory demyelinating polyneuropathy pathogenesis.
- It is not clear to me how did authors connect the Immune reconstitution inflammatory syndrome (IRIS) to the development of CIDP when the patient did not report any history before the initiation of ART , or present with any symptoms or signs of Mycobacterium avium infection after the initiation of ART.
- “Previously undiagnosed MAC infection (unmasking MAC-related IRIS)” how can you confirm the diagnosis of MAC, please discuss.
- Can you please further discuss the Brain MRI signs in figure1?
- What are other possible causes of the presented case (Differential diagnosis), and how did you exclude them?
Author Response
We are deeply honored by the effort you spent in reviewing this manuscript and greatly appreciate your comments. We are motivated to read and learn more from your criticisms in reviewing and revising our text. Our point-by-point responses to your comments are listed below and highlighted the changes in the manuscript (red color).

Reviewer 2 Report
Manuscript refers to case report of Mycobacterium avium complex infection related immune reconstitution inflammatory syndrome (IRIS) presented as chronic inflammatory demyelinating polyneuropathy (CIDP) in young AIDS patient. Manuscript is well written. Authors presented relevant clinical and laboratory findings. Also, they made assumption, based on literature, about mechanisms underlaying disorder shown.
Here are suggestions for improvement of paper.
- Authors should present graphically, on one line graph diagram, kinetics of change of parameters (CD4, CD8, viral load, eventually CD8/CD4 ratio) in different time points - it will be easier for tracking (look at: Narendran G et al. Multifocal tuberculosis-associated immune reconstitution inflammatory syndrome – a case report of a complicated scenario. org/10.1186/s12879-019-4182-1).
- Wheather some autoantibodies against peripheral nerve antigens were determined in the serum of the patient? If not, is there possibility to determine and present results? Supplement discusion with the names of the peripehral nerve antigens (line 153).
- Line 79: What does it means „Chest radiography showed no new infiltration”? Wheather there have been infiltrations before?
- In the abstract, full name of MAC should be written at the place where this abbreviation is used for the first time.
- Line 50: CD8/CD4 instead of CD8:CD4 (in order to be uniformed).
- Line 50: Check the CD8/CD4 ratio. Isn`t it 73,5 instead of 100 (294/4=73,5).
- Line 105 and 184: delete the full stop after the cited references.
Author Response

(The authors gave the same response as above.)
